# Edible Flowers of *Tagetes erecta* L. as Functional Ingredients: Phenolic Composition, Antioxidant and Protective Effects on *Caenorhabditis elegans*

**DOI:** 10.3390/nu10122002

**Published:** 2018-12-18

**Authors:** Cristina Moliner, Lillian Barros, Maria Inês Dias, Víctor López, Elisa Langa, Isabel C.F.R. Ferreira, Carlota Gómez-Rincón

**Affiliations:** 1Department of Pharmacy, Faculty of Health Sciences, Universidad San Jorge, 50830 Villanueva de Gállego (Zaragoza), Spain; acmoliner@usj.es (C.M.); ilopez@usj.es (V.L.); elanga@usj.es (E.L.); 2Centro de Investigação de Montanha (CIMO), Instituto Politécnico de Bragança, Campus de Santa Apolónia, 5300-253 Bragança, Portugal; lillian@ipb.pt (L.B.); maria.ines@ipb.pt (M.I.D.); 3Instituto Agroalimentario de Aragón-IA2 (CITA-Universidad de Zaragoza), 50013 Zaragoza, Spain

**Keywords:** African marigold, edible flowers, polyphenols, antioxidant, neuroprotective potential, *Caenorhabditis elegans*

## Abstract

*Tagetes erecta* L. has long been consumed for culinary and medicinal purposes in different countries. The aim of this study was to explore the potential benefits from two cultivars of *T. erecta* related to its polyphenolic profile as well as antioxidant and anti-aging properties. The phenolic composition was analyzed by LC-DAD-ESI/MSn. Folin-Ciocalteu, DPPH**^·^**, and FRAP assays were performed in order to evaluate reducing antiradical properties. The neuroprotective potential was evaluated using the enzymes acetylcholinesterase and monoamine oxidase. *Caenorhabditis elegans* was used as an in vivo model to assess extract toxicity, antioxidant activity, delayed aging, and reduced β-amyloid toxicity. Both extracts showed similar phenolic profiles and bioactivities. The main polyphenols found were laricitin and its glycosides. No acute toxicity was detected for extracts in the *C. elegans* model. *T. erecta* flower extracts showed promising antioxidant and neuroprotective properties in the different tested models. Hence, these results may add some information supporting the possibilities of using these plants as functional foods and/or as nutraceutical ingredients.

## 1. Introduction

Flowers have been consumed for centuries, but, currently, there is a renewed interest for them not only because they add an aesthetic value to the dishes but also for the healthy properties they may add to the food. The growing interest in edible flowers has increased the research on their nutritional value, biological activities, and bioactive components [1]. The broad range of phytochemicals present in flowers may prevent aging and chronic diseases in relation with inflammation and oxidative stress. There are studies available in the literature about composition and bioactivities of edible flowers. However, more reports related to edible flowers’ safety and usage are needed if they are to be considered functional foods or a part of our diet [2]. The lack of knowledge about their toxicological effect was already highlighted by Egebjerg et al. [3]. They reviewed 23 flowers identified in a control campaign in Danish restaurants and nine of them contained compounds with toxic effects or a potentially toxic effect. In order to avoid these situations, it is necessary to perform studies in relation with phytochemical composition and bioactivities of edible flowers.

*Tagetes erecta* L., which is commonly known as “African marigold” or “Aztec marigold,” is a flowered annual herbaceous plant native from Mexico. This species is widely cultivated for ornamental, poultry, and medicinal purposes [4,5]. Moreover, its flowers are used as an ingredient in salads and as natural food colorant [6] since it is one of the most popular edible flowers all over the world. Most of the previous research is focused on the xanthophyll content and its use as antioxidant for prevention of age-related macular degeneration [7]. However, less attention has been given to phenolic compounds. These circumstances, combined with the great variety of uses, make it important to establish a full profile of composition and biological properties. Some in vitro methods have been employed to assess the antioxidant activity of *T. erecta* extracts regardless of their chemical-biological interactions. To this end, *Caenorhabditis elegans* nematodes have been used.

*C. elegans* is a wide spread model organism to explore the biological effects on aging and human diseases. The nematode genome contains more than 18,000 genes where 60% to 80% are homologue to human ones. Besides the highly conserved metabolic pathways, its easy-to-maintain, short cycle of live, economical, and large collection of mutants are some of the main advantages of the use of *C. elegans* [8,9].

In the present study, two different cultivars of *T. erecta* (yellow and orange flowers) were assessed in terms of the polyphenol profile and antioxidant, anti-aging, and neuroprotective properties. To the best of our knowledge, this is the first time that *T. erecta* flowers have been evaluated in these spectra of bioactivities particularly in the *C. elegans* model.

## 2. Materials and Methods

### 2.1. Standards and Reagents

Acetonitrile (99.9%) was of an HPLC grade from Fisher Scientific (Lisbon, Portugal). Phenolic compound standards (quercetin-3-O-glucoside, myricetin, gallic acid) were from Extrasynthèse (Genay, France). Formic acid, DPPH**^·^** (2,2-diphenyl-1-picrylhydrazyl), TPTZ (2,4,6-tris(2-pyridyl)-s-triazine), ATCI (acetylthiocholine iodide), acetylcholinesterase, monoamine oxidase A (MAO-A), tris-HCl and pyrogallol were purchased from Sigma-Aldrich (St. Louis, MO, USA). DNTB (5,5′-dithiobis (2-nitrobenzoic acid)), lutein, juglone (5-hydroxy-1,4-naphthoquinone), and FUDR (5-fluoro-2′-deoxyuridine) were from Alfa Aesar (Ward Hill, MA, USA). The Folin-Ciocalteu reagent was purchased from Chem-lab (Zeldelgem, Belgium). All other general laboratory reagents were purchased from Panreac Química S.L.U. (Barcelona, Spain). Water was treated in a Milli-Q water purification system (TGI Pure Water Systems, Greenville, SC, USA).

### 2.2. Plant Extracts

Edible flowers of two cultivars of *T. erecta* with yellow and orange petals were purchased from Innoflower SL. Whole fresh flowers were cut into small pieces and extracts were prepared with a soxhlet apparatus using ethanol at an extraction temperature between 80 °C and 85 °C for 4 hours. The solvent was removed with a rotatory evaporator and resulted extracts were stored in the dark at −20 °C.

### 2.3. Analysis of Phenolic Compounds

The phenolic profile was determined in the obtained extracts, which were re-dissolved at a concentration of 10 mg/mL with an ethanol:water (80:20, *v*/*v*) mixture. The analysis was performed by using a LC-DAD-ESI/MSn (Dionex Ultimate 3000 UPLC, Thermo Scientific, San Jose, CA, USA). These compounds were separated and identified as previously described by Bessada et al. [10]. Double online detection was performed using 280, 330, and 370 nm as preferred wavelengths for DAD and in a mass spectrometer (MS). The MS detection was performed in a negative mode using a Linear Ion Trap LTQ XL mass spectrometer (Thermo Finnigan, San Jose, CA, USA) equipped with an ESI source.

The identification of the phenolic compounds was performed based on their chromatographic, UV-Vis, and mass spectra data by comparing it with standard compounds when available data is reported in the literature. Data acquisition was carried out with an Xcalibur® data system (Thermo Finnigan, San Jose, CA, USA). For quantitative analysis, a calibration curve for each available phenolic standard was conducted based on the UV-vis signal. For the identified phenolic compounds for which a commercial standard was not available, the quantification was performed through the calibration curve of the most similar available standard: quercetin-3-*O*-glucoside (*y* = 34843*x* – 160173, *R*^2^ = 0.999), myricetin (*y* = 23287*x* – 581708, *R*^2^ = 0.999), and gallic acid acid (*y* = 131538*x* + 292163, *R*^2^ = 0.999). The results were expressed as mg/g of extract.

### 2.4. In Vitro Antioxidant Activity

#### 2.4.1. Determination of Folin-Ciocalteu Reducing Capacity

The Folin-Ciocalteu reducing capacity was determined according to the literature [11] with minor modifications. A total of 201 µL of Folin-Ciocalteau reagent were mixed with 9 µL of diluted extracts in ethanol (2.5, 5, and 10 µg/mL). After 5 minutes in the dark at room temperature, 90 µL of Na_2_CO_3_ (10%) were added drop by drop. The absorbance of the reaction mixture was measured at 752 nm after incubation for 40 minutes at room temperature in darkness. TPC was determined by comparison with the standard curve of pyrogallol and the results were expressed as mg of pyrogallol equivalent per g of extract (mg PE/g extract).

#### 2.4.2. Radical Scavenging Activity (RSA)

DPPH radical-scavenging activity of the extracts was evaluated through spectrophotometric techniques following the method previously described [12]. A total of 150 μL of different dilutions of the extracts in ethanol were mixed with 150 μL of a DPPH**^·^** ethanol solution (0.04 mg/mL) and incubated in the dark at room temperature for 30 minutes. The absorbance values were measured at 518 nm and converted into the Radical Scavening Activity, % RSA, calculated as %RSA = ((Abs_control_ – Abs_sample_)/Abs_control_) × 100 where the Abs_control_ is the absorbance of DPPH**^·^** without extract (control) and the Abs_sample_ is the absorbance of DPPH**^·^** with the extracts.

#### 2.4.3. Ferric Reducing Antioxidant Power (FRAP) Assay

This assay was performed according to the procedure described in the literature with minor modifications [13]. The FRAP reagent was prepared daily and contained TPTZ (10 mmol/L in 40 mmol/L HCl), FeCl_3_·6H_2_O (20 mmol/L), and sodium acetate buffer (300 mmol/L, pH 3.6) in a volume ratio 1:1:10, respectively. Then, 900 μL of FRAP reagent with 90 μL distilled water and 30 μL of flower extracts (1mg/mL) were mixed. After 30 minutes at 37 °C, the absorbance was measured at 595 nm. The standard curve was prepared with FeSO_4_·7H_2_O and the reducing power was expressed as μmol Fe^2+^/g extract.

### 2.5. Neuroprotective Potential

#### 2.5.1. Inhibition of the Acetylcholinesterase Enzyme (AChE)

Inhibitory AChE activity of the extracts was evaluated using Ellman´s method adapted to 96 well microplate [14]. Additionally, 50 µL of buffer (50 mmol/L Tris-HCl, pH 8, 0.1% bovine serum), 25 µL acetylcholine iodide, 25 µL of 3 mM Ellman´s reagent, and 25 µL of extracts dilutions in ethanol were mixed in each well. Lastly, 25 µL of AChE (0.22 U/mL) were added. The percentage of enzyme inhibition, % EI, was calculated from the measurement of absorbance at 405 nm every 12 s five times using Equation (1).
% EI = (1 − (Vsample/Vcontrol)) × 100(1)
where the Vsample is the reaction rate of the extract and Vcontrol is the reaction rate in the absence of the sample.

#### 2.5.2. Inhibition of Monoamine Oxidase A (MAO-A) Enzyme

The inhibition of MAO-A activity was assessed by the peroxidase-linked assay previously described [15]. Each test contained 50 µL flower extract or solvent, 50 µL of chromogenic solution (0.8 mM vanillic acid, 417 mM 4-aminoantipyrine, and 4 U/mL horseradish peroxidase), 100 µL 2.5 mM tryramine, and 50 µL 8 U/mL MAO-A. All components were dissolved in potassium phosphate buffer (0.2 M, pH 7.6). The absorbance was measured at 490 mn every 5 minutes for 30 minutes. The percentage of enzymatic inhibition was calculated using Equation (1).

### 2.6. C. elegans Assays

#### 2.6.1. *C. elegans* Strains and Maintenance Conditions

The strains used in this study were N2 (wild-type) and CL4176 (smg-1ts 131 (myo-3/Aβ1–42 long 3′-UTR)). All strains and *Escherichia coli* (OP50-uracil auxotorph) were obtained from the Caenorhabditis Genetics Center (CGC, Minneapolis, MN, USA). *C. elegans* N2 was maintained and assayed at 20 °C on Nematode Growth Medium (NGM) seeded with *E. coli* and the temperature sensitive strain CL4176 at 16 °C. Synchronized worms N2 were obtained by an alkali-bleaching method [16] and for strain CL4176 by egg-laying.

#### 2.6.2. Assessment of Acute Toxicology

Toxicity tests in liquid medium were performed as described by Donkin and Williams with modifications [17]. Synchronized populations were allowed to develop until larva stage 4 in NGM plates at 20 °C. Afterward, the nematodes were washed off twice with sterile water and re-suspended in K-medium (32 mM KCl, 51 mM NaCl) at a concentration of 80–120 worms/mL. 200 µL of this suspension were mixed in a well with 50 µL of flower extract or K-medium as a negative control. The nematodes were exposed to seven concentrations of extracts in the range of 50–2000 μg/mL using 40 individuals per treatment. The number of dead worms was counted and recorded after 24 hours. The results were expressed as a percentage of survival rate or viability:% Survival rate = (Number of alive worms × 100)/Total number of worms(2)

#### 2.6.3. Evaluation of Resistance to Oxidative Stress

Synchronized L1 worms were transferred to NGM agar plates containing different concentrations of flower extracts (62.5, 125, and 250 μg/mL) or in the absence of them and cultivated at 20 °C until the first day of adulthood. After the exposure period, the worms were washed twice with sterile water. Then, adults were transferred to a 96-well microplate with NGM agar containing 150 μM juglone, which produces lethal oxidative stress, and was cultivated for 24 hours at 20 °C [18]. The lack of touch response with a platinum wire was used to score them as died. The result is presented as a percentage of survival rate (Equation (2)). At least 120 worms per condition were evaluated in each assay.

#### 2.6.4. Lifespan Assay.

This assay has been carried out following the method of Solis and Petrascheck with a slight modification [19]. Synchronized worms (L1) were transferred to 96-well plates (7–18 worms/well) and cultured in S-complete medium containing *E. coli* OP50 (1.2 × 10^9^ bacteria/mL) as feeding bacteria. On the first day of adulthood, FUDR (0.06 mM) was added to sterilize animals. After 24 hours, different concentrations of extracts (50, 75, 125, and 250 μg/mL) were added. Life animals were scored as dead in the absence of movement and recorded three times a week. Each condition was performed in at least 125 nematodes.

#### 2.6.5. Paralysis Assay

The *C. elegans* strain CL4176 contains a temperature sensitive mutation that expresses human amyloid β1–42. The Aβ peptides are expressed and aggregated in the muscle cells, which causes paralysis in the mutants. The strain CL4176 was egg-synchronized onto the NGM plates seeded with *E. coli* in the presence of a range of concentrations of the extracts (50–250 μg/mL) and in the absence of them at 16 °C. In addition, 38 hours after the egg laying, there was an upshifting of the temperature to 25 °C to induce Aβ transgene expression. The paralysis was scored 24 h after the initiation of upshift until 72 h [20]. Worms were considered as paralyzed or dead after failure to complete one sinusoidal turn within 5 s of being touched on the head and tail with a platinum wire. For each assay, at least 100 nematodes were studied per condition.

### 2.7. Statistical Analysis

GraphPad Prism version 6.0c for Mac OS X (GraphPad Software, San Diego, CA, USA) was used for statistical analysis. All experiments were performed in triplicate and their results were plotted as mean ± standard error means (SEM). In the DPPH**^·^**, MAO A, and AchE inhibition assay, IC_50_ values (concentration of extract required to scavenge 50% DPPH**^·^** or inhibit the enzyme) were estimated by using a non-linear regression. Evaluation of acute toxicology and resistance to oxidative stress was analyzed by using the ANOVA and Tukey’s multiple comparisons test. Lifespan and paralysis curves were analyzed by Kaplan–Meier survival curves and by conducting log-rank tests. Differences with *p* ≤ 0.05 were considered statistically significant.

## 3. Results and Discussion

### 3.1. Identification and Quantification of Phenolic Compounds of T. erecta Extracts

Extracts were prepared from fresh flowers of two cultivars of *T. erecta* with different inflorescence pigmentation (orange and yellow). Yields for the two plants were 3.17% (mass of extract/ mass of fresh flowers) for orange flowers and 3.45% for the yellow ones. Both samples were analyzed by using LC-DAD-ESI/MSn and they had a similar phenolic profile, which is summarized in Table 1. In total, nine compounds were identified in the ethanolic extracts including one phenolic acid (digallic acid) and eight flavonols (laricitrin and myricetin derivatives). A representative chromatogram of the ethanolic extracts recorded at 280 and 370 nm is shown in Figure 1. Only one phenolic acid was identified in the ethanolic extracts due to being tentatively identified as digallic acid (peak 1) presenting a pseudomolecular ion [M-H]^−^ at *m*/*z* 321 with an MS^2^ fragment at *m*/*z* 169 (gallic acid molecule) corresponding to the loss of the 152 u of a galloyl group.

Regarding the flavonoids group, laricitrin was the most representative aglycone in the ethanolic extracts, which is in accordance with a previously performed study by Navarro-González et al. in an acidified ethanolic extracts of *T. erecta* from Spain [6]. The aglycone laricitrin (peak 9) was tentatively identified with the pseudomolecular ion [M-H]^−^ at *m*/*z* 331 and the characteristic fragmentation pattern with MS^3^ fragments at *m*/*z* 316, 287, and 271. Peaks 6, 7, and 8 ([M-H]^−^ at *m*/*z* 493) and peaks 2 and 4 ([M-H]^−^ at *m*/*z* 655) presented a unique MS^2^ fragment at *m*/*z* 331 corresponding to the loss of one ([M-162]^−^) and two ([M-162-162]^−^) hexosyl moieties, which are tentatively identified as laricitrin-hexoside and laricitrin-di-hexoside, respectively. Peak 5 was tentatively identified as laricitrin-galloyl-hexoside presenting a pseudomolecular ion [M-H]^−^ at *m*/*z* 645 and two main MS^2^ fragments at *m*/*z* 493 (−152 u) and 331 (−162 u, laricitrin algycone) corresponding to the loss of a galloyl and hexosyl moieties. Lastly, peak 3 was identified as a myricetin derivative, which presents a pseudomolecular ion [M-H]^−^ at *m*/*z* 479 and a unique MS^2^ fragment at *m*/*z* 317 (−162 u, hexosyl residue). This is tentatively assigned to a myricetin-hexoside.

For the orange and yellow cultivars, peaks 6 (laricitrin-hexoside), 3 (myricetin-hexoside), and 9 (laricitrin) were the major compounds found. No significant differences in the total concentration of the phenolic compounds between both cultivars were presented, which quantified a total phenolic content of 51.06 ± 0.24 and 54.08 ± 0.4 mg/g of extract for the orange and yellow flowers, respectively. Similarly, no significant differences were detected when both groups of compounds, phenolic acids, and flavonoids were compared including the phenolic acid content of 1.23 ± 0.04 mg/g of extract vs. 1.25 ± 0.02 mg/g of extract and flavonoids 49.8 ± 0.2 mg/g of extract vs. 53.5 ± 0.5 mg/g of extract. The major components including laricitrin and its glycosides have previously been identified in the *T. erecta* by other authors [6] who also reported the presence of the myricetin derivative and tannins. Moreover, Kaisoon et al. revealed the presence of other phenolic acids and flavonoids [5]. These differences could be related to the different origin of the flowers or the extraction procedure applied to extract these compounds.

### 3.2. In vitro Antioxidant Activity

In order to confirm antioxidant properties of the extracts, three different methods have been used such as Folin-Ciocalteu, DPPH**^·^** radical scavenging, and FRAP assays [21]. The results are given in Table 2.

Reducing capacity of the extracts was quantified by the Folin-Ciocalteu method, which is based on an oxidation reaction, and values were expressed in terms of mg of pyrogallol equivalent (PE) per g of extract. This method has been also used as an estimation of total polyphenol content. Both extracts showed a similar content of 77 ± 3 mg of PE/g of extract for orange flowers and 81 ± 3 mg of PE/g of extract for the yellow one. These values are higher than those obtained with LC-DAD-ESI/MSn. Overestimation of total phenolic content by Folin-Ciocalteu is due to non-specificity of the method as non-phenolic compounds such as vitamins or proteins, which react towards the Folin-Ciocalteu reagent. However, this assay is still a recognized method to assess the total antioxidant-reducing activity [22]. Generally speaking, flowers of *T. erecta* showed a great variability regarding the phenolic content. For example, some studies reported higher TPC values than our data [5,23] while others showed lower ones [6,24] and others found values in the same range [25]. The study carried out by Li et al. deserves a particular mention [23]. In the cited work, the phenolic content of 11 cultivars of *T. erecta* flowers was analyzed and huge variations between them were found. Hence, the variability among same species, the use of different extraction solvent and standards generate differences, which makes it difficult to compare data.

The maximum percentage of inhibition of DPPH**^·^** was 87.0% ± 0.6% for orange flowers and 86.9% ± 0.6% for yellow ones with similar IC_50_ values (Table 2). Li et al. also studied the DPPH**^·^** inhibition of methanolic extracts of 11 *T. erecta* flower cultivars [23]. Kaisoon et al. found similar results [26].

In the FRAP assay, the orange cultivar showed a significantly higher reducing power than a yellow one (112 ± 3 µmol Fe^2+^/g extract vs. 78 ± 7 µmol Fe^2+^/g extract, respectively). Differences in FRAP value could be associated with the compounds responsible for the color of petals. These pigments, which are xanthophylls and carotenes [27], have previously shown ferric reducing activity [28]. This fact might not affect the DPPH**^·^** values because carotenoids are not able to scavenge the DPPH**^·^** due to the presence of hydroxyl and keto functional groups on the terminal rings [28,29].

Scientific literature reveals strong scavenging and reducing power for these species, which is highlighted in our work, when compared with the range reported for several edible flowers [24,26].

### 3.3. Neuroprotective Potential

Ethanolic extracts of fresh flowers were tested for their capacity to inhibit AChE and MAO-A using in vitro procedures. Both extracts inhibited these enzymes in a clear dose-dependent manner. IC_50_ values of AChE were 1.13 ± 0.05 mg/mL for orange flowers and 1.25 ± 0.07 mg/mL for yellow ones (Table 2). To the best of our knowledge, this is the first report about inhibitory activity in these enzymes. Inhibition of these physiological targets may be in relation with mental health and neuroprotective potential. They may act by modulating the breakdown of certain neurotransmitters and this fact might explain the mechanism of action that supports the traditional use of these plant species in Mexico, reported by Pérez-Ortega et al. [30].

### 3.4. Evaluation of T. erecta Extracts Acute Toxicity

The acute toxicity of the extracts was assessed by studying their impact in the viability of *C. elegans*, which is a model widely used in the field of toxicology due to similarities between nematodes and mammalians [31]. The treatment with extracts in a range of concentrations between 50 and 2000 μg/mL for 24 h did not affect worm viability when compared to the control group (Appendix A). Even at a maximum dose (2000 μg/mL), worms maintained a viability of 91% ± 2% for orange flowers and 96% ± 1% for yellow ones while the viability rates of control groups were 90% ± 2 % and 93% ± 2%, respectively.

### 3.5. T. erecta Exerts Protective Effects on C. elegans under Lethal Oxidative Stress

Oxidative stress was induced on *C. elegans* by exposure to a lethal dose of juglone, which is a strong natural pro-oxidant. Pre-treatments with the extracts had a positive impact on the response of nematodes under oxidative stress by increasing the survival rate in relation with a control group (Figure 2). The rates of survival were significantly higher at doses of 125 and 250 μg/mL for orange flowers and at all tested concentrations for yellow flowers (*p* < 0.05). The best response to oxidative stress was found in the group treated with 250 µg/mL of both extracts for which the survival rate was increased from 4% ± 1% (control group) to 28% ± 5% (250 μg/mL) in orange cultivar while, in the yellow one, was from 2.8% ± 0.8%(control group) to 21% ± 3% (250 μg/mL). Our results clearly indicate that ethanolic extracts of *T. erecta* flowers protect *C. elegans* from oxidative stress. The response to oxidative stress due to the extracts could be attributed to laricitrin and its glycosides, which are abundant in the extract and increased the stress resistance of the nematode [32]. In addition, myricetin and other flavonoids present at minor quantities has shown to reduce oxidative damage of biomolecules [33]. On the other hand, the carotenoid lutein has been considered one of the responsible compounds for antioxidant activity of *T. erecta* extracts [25]. For this reason, the same assay was carried out with lutein in a range of 1–100 μM and no protective effects were found (data not shown). Hence, lutein has no impact on a response to oxidative stress induced by juglone in *C. elegans*. By contrast, Augusti et al. pointed out that low concentrations of lutein could mitigate the oxidative stress in *C. elegans* induced by microcystin-LR [34].

### 3.6. T. erecta Flower Extracts Enhance the Lifespan of C. elegans

In order to determine the life extension properties of extracts, a lifespan assay in liquid medium was performed at 20 °C using wild-type worms, which were exposed to different concentrations of extracts. Figure 3 shows the survival curves of the 250 µg/mL treatment groups against the control group plotted by the Kaplan-Meier model. The results revealed a significantly increase of the lifetime of treated worms. A concentration-dependent effect was observed for both extracts compared to the control group. For orange cultivar, all concentrations significantly prolonged the lifespan of the nematode. However, in yellow cultivar, the lowest concentration tested was 50 µg/mL and did not affect the lifespan, while, in all other treatments, it had a significantly positive impact on the length of life. The best concentration prolonging the lifespan was 250 µg/mL for both extracts. An extension from 18 days (control group) to 22 days was detected for orange flowers and from 15 days (control group) to 18 days for yellow cultivar. The protective effect of *T. erecta* flowers on delaying aging becomes evident in these experiments (Figure 3). The mechanism of life enhanced *T. erecta* flowers extracts, which has not been elucidated yet. However, similar experiments were carried out with the major phenolic compounds present in the extracts laricitrin and myricetin, which showed a lifespan extension in a DAF-16-dependent manner [32,33]. DAF-16 is a transcription factor known as the FOXO homologue in *C. elegans*. FOXO proteins are linked to longevity in humans [35] and also promote neuronal health [36].

### 3.7. Extracts of T. erecta Flower Delayed the Paralysis of β-Amyloid (Aβ) Transgenic C. elegans

The β-amyloid (Aβ) peptide is well-known as a neurotoxic agent. The CL4176 strain has proven to be an interesting model in *C. elegans* for the screening of neuroprotective bioactives [37]. To this end, CL4176 larvae were treated with different concentrations of the extracts (50, 100, and 250 µg/mL) at 16 °C for 48 hours. Then, human Aβ expression was induced by a temperature upshift, which led to Aβ deposition in the muscles cells. This causes the paralysis of the nematodes. The paralysis curves are shown in Figure 4 from which the time when 50% of worms were paralyzed or dead, PT_50_, was calculated. A statically significance dose-dependent manner was shown in the paralysis curves of both extracts. Compared with the control, the orange cultivar increased the mean time of developed paralysis of CL4176 worms in approximately 30% in all treatments. On the other hand, the yellow flower extract delayed significantly the PT_50_ by around 11% at 50 µg/mL, 33% at 100 µg/mL, and, for the highest dose tested, the PT_50_ was not reached. These findings demonstrate the neuroprotective effect of flowers of *T. erecta* in the *C. elegans* AD model.

On the basis of the obtained results, the protection against Aβ accumulation could be due to a dual mechanism involving the inhibitory activity of CNS enzymes and antioxidant properties. MAO and AChE inhibitors have an important role in the treatment of neurodegenerative disorders such as Alzheimer’s and Parkinson’s disease, slowing their progression [38]. Both enzymes are also present in the neurotransmitter metabolism of *C. elegans* [39] and their inhibition could delay the onset of paralysis on the transgenic nematode. Moreover, an accumulation of oxidative damage precedes the appearance of AD symptoms [40]. This feature is also included in the AD model used where an oxidative stress condition before the Aβ deposition was observed [41]. Another data point to consider in the assessment of the neuroprotective effects of *T. erecta* extracts is the impact of myricetin to reduce the production of β amyloid protein in neuronal cultures [42]. These results are in concordance with the rich phenolic content and the antioxidant potential of the studied extracts.

The use of functional foods has a great potential for reducing the risk of age-related diseases. However, plant foods need evidence of their positive health benefits to be considered as functional foods or nutraceuticals. The difficulty in performing clinical trials to prove their efficacy in disease conditions makes it necessary to be backed with in vitro and in vivo models [8,43]. This evidence supports the need to continue with research on natural plant ingredients and particularly edible flowers as new sources of polyphenols and bioactive compounds, which may display positive effects on health.

## 4. Conclusions

This work confirms that flowers of *T. erecta* contain flavonols such as laricitrin and its glycosides. The two different cultivars studied are yellow and orange petals and they showed a similar phenolic profile and in vitro antioxidant properties supported with protective in vivo effects on *C. elegans*. The extracts had a neuroprotective potential, as shown by the ability to inhibit CNS enzymes and mitigate the effect of β-amyloid toxicity on a transgenic strain of *C. elegans*. Overall, this study promotes more research about the use of *T. erecta* flowers as a dietary source of bioactive compounds.

## Figures and Tables

**Figure 1 nutrients-10-02002-f001:**
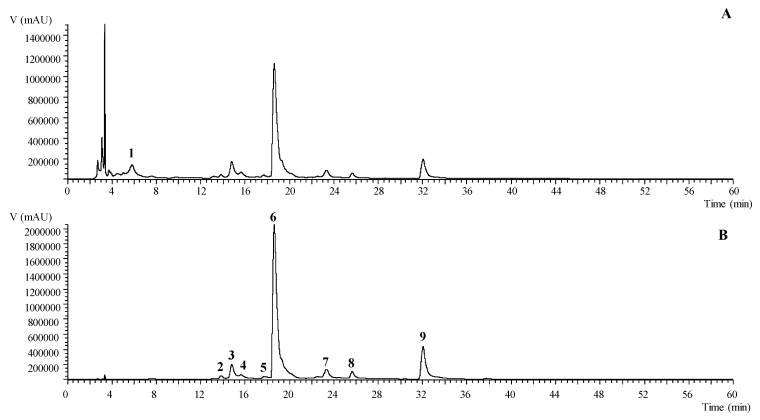
Phenolic profile of *T. erecta* yellow recorded at 280 nm (**A**) and 370 nm (**B**).

**Figure 2 nutrients-10-02002-f002:**
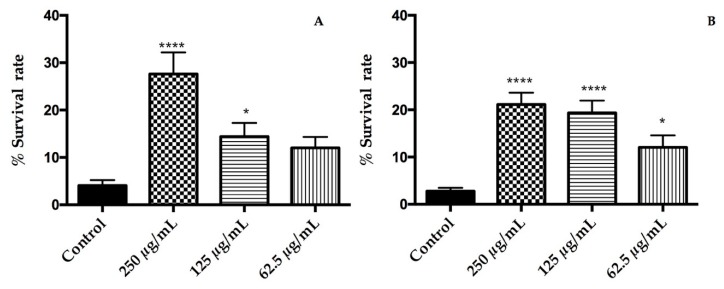
Effects of (**A**) orange cultivar and (**B**) yellow cultivar of *T. erecta* flowers extracts on the response to a lethal oxidative stress induced by juglone on *C. elegans*. Differences compared to the control group were considered significant at *p* < 0.05 (*) and *p* < 0.0001 (****).

**Figure 3 nutrients-10-02002-f003:**
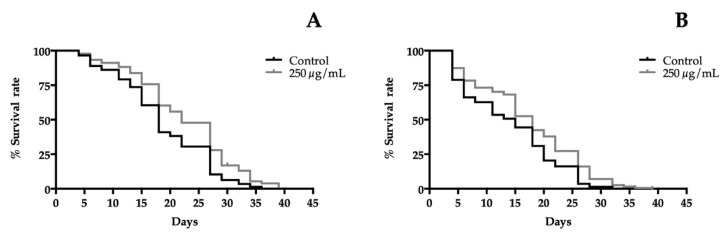
Effect of the highest dose tested of (**A**) orange cultivar and (**B**) yellow cultivar of *T. erecta* flowers extracts on lifespan of wild-type *C. elegans*. The mean of the lifespan for orange cultivar were: 18 days (control group), 20 days (50, 75, and 125 µg/mL treated groups) and 22 days (250 µg/mL treated group). For yellow cultivar were: 15 days in all groups except in nematodes treated with 250 µg/mL, which was 18 days. The results of lifespan experiments were analyzed by using the Kaplan-Meier survival model and for significance by using a long rank pairwise comparison test between the control and treatment groups. Differences in survival curves between treatment and control groups were found in: (**A**) 50 *, 75 ***, 125**, and 250 **** µg/mL. (**B**): 75 *, 125 *, and 250 *** µg/mL. Differences compared to the control group were considered significant at *p* < 0.05 (*), *p* < 0.01(**), *p* < 0.001 (***), and *p* < 0.0001 (****).

**Figure 4 nutrients-10-02002-f004:**
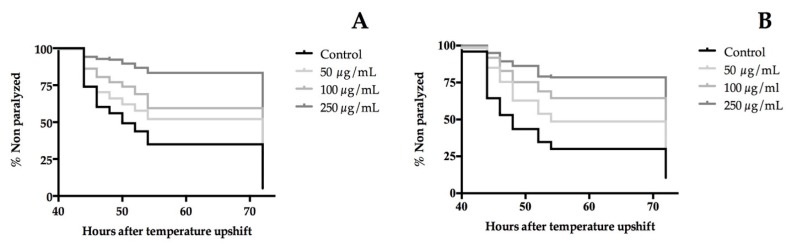
Effect of (**A**) orange cultivar and (**B**) yellow cultivar of *T. erecta* flower extracts on Aβ-induced paralysis on transgenic *C. elegans* CL4176. The PT_50_ for orange cultivar were 50 hours for the control group and 72 hours for all treated groups and for yellow cultivar were 48 hours (control group), 54 hours (50 µg/mL treated group), 72 hours (100 µg/mL treated group), and the PT_50_ was not achieved in the group treated with 250 µg/mL. Statistical significance of the difference between the experiments was analyzed by a log-rank (Kaplan-Meier) statistical test, which compares the survival distributions between the control and treated groups. Differences in survival tests between treatment and the control group were found in: (**A**) 50, 100, and 250 µg/mL, *p* < 0.0001, (**B**) 50, 100, and 250 µg/mL, *p* < 0.0001.

**Table 1 nutrients-10-02002-t001:** Retention time (Rt), wavelengths of maximum absorption in the visible region (λ, w), mass spectral data, tentative identification, and quantification (mg/g of extract) of the phenolic compounds present in orange and yellow cultivars of *Tagetes erecta* L. flowers.

Peak	Rt (min)	λ_max_ (nm)	Molecular ion [M-H]^−^ (*m*/*z*)	MS^2^ (*m*/*z*)	Tentative Identification	Quantification		*t*-Students Test *p*-Value
						*T. erecta Orange*	*T. erecta Yellow*	
1	5.81	273	321	169 (100)	Digallic acid ^A^	1.233 ± 0.004	1.25 ± 0.02	0.007
2	13.83	360	655	493 (100), 331 (11)	Laricitrin-di-hexoside ^B^	1.669 ± 0.001	1.477 ± 0.003	0.110
3	14.80	359	479	317 (100)	Myricetin-hexoside ^C^	4.42 ± 0.03	3.6 ± 0.1	0.008
4	15.64	354	655	493 (45), 331 (100)	Laricitrin-di-hexoside ^B^	2.53 ± 0.01	1.859 ± 0.003	0.016
5	17.67	356	645	493 (100), 331 (16)	Laricitrin-galloyl-hexoside ^B^	1.444 ±0.001	1.466 ± 0.001	0.116
6	18.63	370	493	331 (100)	Laricitrin-hexoside ^B^	28.9 ± 0.1	31.5 ± 0.4	0.40
7	23.33	362	493	331 (100)	Laricitrin-hexoside ^B^	3.69 ± 0.03	3.20 ± 0.04	0.261
8	25.67	365	493	331 (100)	Laricitrin-hexoside ^B^	3.02 ± 0.03	2.328 ± 0.001	0.01
9	32.06	368	331	316 (100), 287 (5), 271 (5)	Laricitrin ^B^	4.1 ± 0.1	8.08 ± 0.01	0.02
**Total Phenolic Acids**						1.23 ± 0.04	1.25 ± 0.02	0.07
**Total Flavonoids**						49.8 ± 0.2	53.5 ± 0.5	0.131
**Total Phenolic Compounds**						51.1 ± 0.2	54.8 ± 0.4	0.145

Standard calibration curves: A—gallic acid (*y* = 131538*x* + 292163, *R*^2^ = 0.999), B—quercetin-3-O-glucoside (*y* = 34843*x* − 160173, *R*^2^ = 0.999), C—myricetin (*y* = 23287*x* − 581708, *R*^2^ = 0.999), *p*-values were calculated in order to detect significant differences between the two cultivars, *p* values > 0.05 indicate significant differences.

**Table 2 nutrients-10-02002-t002:** Antioxidant and enzyme inhibition activity of flower extracts. The results are presented as mean ± SEM.

	Folin-Ciocalteu mg PE/g Extract	DPPH^·^ IC_50_ (µg/mL)	FRAP µmol Fe^2+^/g Extract	AchE IC_50_ (mg/ML)	MAO-A IC_50_ (mg/mL)
Orange cultivar	77 ± 3	21.3 ± 0.9	112 ± 3	1.13 ± 0.05	0.023 ± 0.001
Yellow cultivar	81 ± 3	22.0 ± 1.0	78 ± 7	1.25 ± 0.07	0.024 ± 0.003
*t*-Studens test *p*-value	>0.05	>0.05	≤0.05	>0.05	>0.05

*p*-values were calculated to detect differences between the two cultivars, *p* values > 0.05 indicate significant differences.

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
