# Peer review of "Edible Flowers of Tagetes erecta L. as Functional Ingredients: Phenolic Composition, Antioxidant and Protective Effects on Caenorhabditis elegans"

_nutrients, 2018, doi:10.3390/nu10122002_

Round 1
Reviewer 1 Report
On the basis of careful study of the manuscript number 395660, it results well written, the experiments well designed and the topic relevant to be published in Nutrients.
Only few comments:
The authors have measured the in vitro antioxidant activity with Folin-Ciocalteu:
Have they some explanation about this high variability?
Further, the reported neuroprotective potential activity is very interesting, have the authors information if the used concentrations can have some possible in vivo effect even considering the low bioavailability of polyphenols?
Author Response
Reviewer 1:
Only few comments:
Point1: The authors have measured the in vitro antioxidant activity with Folin-Ciocalteu:
Have they some explanation about this high variability?
Answer1: The variation coefficient is less than 10% which can be considered an acceptable value. Moreover, the results shown in Table 2 for the Folin-Ciocalteu test were obtained in three independent experiments conducted under identical conditions on three different days in order to decrease the variability associated with the methodology. On the other hand, other authors (González-Barrio et al., 2018; Koike et al., 2015) obtain similar or even higher variability values. This suggests that such variability can be considered inherent to the method itself. However, this method is considered useful in determining the reducing power.
Point 2: Further, the reported neuroprotective potential activity is very interesting, have the authors information if the used concentrations can have some possible in vivo effect even considering the low bioavailability of polyphenols?
Answer 2: As suggested, polyphenols, like other potentially neurobeneficial compounds, may have low bioavailability in in vivo models. However, different authors have shown the capacity of some polyphenols to cross the blood-brain barrier in vitro (Kuresh et al., 2004) as well as their neuroprotective efficacy in vivo in murine models (Jeong et al., 2018) and in clinical trials (Mastroiacovo et al., 2014). Although our results are promising, new studies in higher animals and clinical trials would be necessary to extrapolate these effects in humans.
Reviewer 2 Report
This study presents the results on qualitative and quantitative composition of some phytochemicals (mainly flavonoids) in ethanol extracts of fresh marigold flowers (2 cultivars) and the activities of these extracts using several in vitro assays and Caenorhabditis elegans nematode. To my opinion, the value some of these results (phytochemical composition, antioxidant activity) is rather low due to the negligible scientific novelty; while the results with the tested enzymes and worms provide new information on the bioactivities of marigold extract. Therefore, the manuscript, including introduction and discussion, should be better focused by emphasizing new and most important findings. The Authors should consider that numerous articles on T. erecta have been published until now; therefore proving the novelty of the reported findings and their importance should be essential and convincingly shown. The manuscript also contains quite many inaccuracies, some speculative and unclear statements. They are provided below.
Line 13 (and elsewhere): Tagetes erecta is also known as Mexican marigold or Aztec marigold (not only an African marigold); therefore, all common names should be given or, preferably, only botanical name should be left in the abstract.
Line 16: Folin = Folin-Ciocalteu
Line 23-24: To my opinion, the results of this study do not support the claim ‘T. erecta flowers might be used as functional foods”. More correctly, these results may add some information supporting the possibilities of using this plant for functional food and/or nutraceutical ingredients.
Line 25: To my opinion, ‘functional foods’ is too general and not sufficiently relevant keyword for this study.
Line 34: I would not agree with the statement: ‘there are few available studies’. For instance, search word ‘Tagetes erecta’ provides a large number of publications in Clavirate Analytics WoS database: Tagetes erecta+antioxidant =80; Tagetes erecta+phenolic composition = 17; Tagetes erecta+protective = 17. In total, thousands of records may be found for different flowers in terms of their composition and bioactivities.
Line 64: ‘Formic acid DPPH”? Most likely, formic acid and DPPH. In addition, DPPH is a stable radical and it should be indicated by the word or, preferably, by the superscript dot.
Line 74: Plant variety is a legal term, following the International Union for the Protection of New Varieties of Plants (UPOV) Convention; in such case more detailed information should be provided for the varieties, e.g. Tagetes erecta L., var? Or these were just different cultivars (accessions)?
Line 75: How did the Authors ensure that extraction was sufficient for the recovery of bioactive compounds?
Line 76: How the temperature of 110 C was reached in a Soxhlet using ethanol (boiling point = 78 C)?
Line 76: Fresh flowers contain large portion of water, which is miscible with ethanol. What was the content of remaining water after solvent removal or water was also evaporated? How it was assessed?
Line 89: ‘behaviour’ = data.
Line 86: The yield of the extract (e.g. from 1 kg of fresh or dried material) in such study would be very important for characterizing the potential of the plant. For instance, the extract may be strong antioxidant but when its yield is very low, the prospects of commercial plant application may be low due to economic issues.
Line 100: What was the reason of selecting this article for the citation? ‘Folin-Cicolteau’ = Folin-Ciocalteu.
Line 101-102: The same as in line 81?
Line 102: ‘Na2CO3’ = Na2CO3.
Line 105: What was the reason of using pyrogallol, when gallic acid was available?
Line 109: ‘The radical scavenging activity (RSA) or’ may be deleted.
Line 110: This is confusing: DPPH radicals are not reduced ‘through spectrophotometric techniques’ (should be rephrased).
Line 111 (and elsewhere): ‘0,04’ = 0.04
Lines 114-117: This is rather trivial and may be replaced by the simple explanation in the text.
Line 125: ‘FeSO4·7H20’ = FeSO4·7H20
Lines 159, 178 (and elsewhere): Citation style: Is it correct?
Line 180: E. Coli = E. coli
Line 190: E. coli should be in italics
Data in section 3.1. From the scientific novelty point of view the value of this data is rather low; it just repeat previously reported information on the composition of phenolics in T. erecta (reporting the composition of other samples is just a technical work and provides very little novelty). There is also a question regarding the accuracy of MS (also in Table 1): the Authors provide m/z of pseudomolecular ions [M-H] and fragments as whole numbers (without any decimals), which indicates that the accuracy was 1 amu. Is that correct? Also standard deviations in Table 1 are unbelievably low for biological material. Therefore, it may be supposed that these values are obtained from 3 replicate chromatographic runs of the same extract, which is not sufficient for the reliable statistical evaluation. For this purpose 3 replicate extractions from the same plant accession had to be performed. In addition, the content of these compounds in the raw plant material was not determined.
Line 263: ‘antioxidant models’ = assays.
Lines 262-264: The sentence should be rephrased.
Line 264: What means ‘appropriate evaluation’? All these assays are based on a single electron transfer (hydrogen atom transfer may also proceed in DPPH assay) and are not relevant to biological systems. The experts strongly recommend using with FC/DPPH/FRAP assays an ORAC assay, which is better linked to the oxidation processes in biological systems.
Lines 268-269 (the title of Table 2): the units are given in the first row of the Table for every assay; it is not necessary to repeat it in the title.
Table 1 and 2: What mean ‘p values’ in these tables? Usually, significant differences between values are indicated by different superscript letters.
Lines 273-304: To my opinion. this part should be substantially shortened; in general antioxidant capacity measured in this study also is of a little novelty for marigold flowers.
Line 308: The values in Table 2 do not provide information on ‘dose-dependent’ character of their inhibitory activity.
Lines 316, 326: Why systematic names in the titles are given using a normal script (not italics)?
Section 3.4: Can these results be compared with: Pina-Vazquez, DM et al.? Anthelmintic effect of Psidium guajava and Tagetes erecta on wild-type and Levamisole-resistant Caenorhabditis elegans strains J. Ethnopharmacol., 202, 92-96 (2017).
Lines 322-324: To my opinion this comparison is not adequate: essential oils are completely different substances; also the cited study was performed with mice.
Line 326: The title should be rephrased.
Lines 332-333: Not clear, when compared with the data in Fig.2 showing that survival rate at the highest concentration was several times higher compared with control. Should be rephrased (not ‘by’ but ‘from….to’).
Line 335: Not clear: ‘The enhanced response to oxidative stress of extracts’.
Line 338: This is a too preliminary and speculative assumption; not supported by this study.
Lines 339-340: This is in disagreement with the statement in lines 298-299.
Lines 343-344: Should be rephrased.
Line 351: ‘of the flowers’ = of extracts.
Line 355: ‘in both extracts’ = for (or ‘in case of’) both extracts.
Lines 358-360: This sentence should be rewritten.
Line 371: To my opinion, this figure may be simplified by leaving only the highest doses; the effects of other doses are shortly explained in the text.
Line 414: ‘therapeutic strategies’? More correctly, ‘disease preventing or reducing their risks’.
Line 415: ‘foods needs’ = foods need.
Line 419: What mean ‘non-nutrient bioactives’? A nutrient is a general term, encompassing both macro and micronutrients.
Line 422: ‘reveals’ or confirms previously reported data?
Line 422: When the concentration is considered as ‘high’ (‘medium’, ‘low’)? Also regarding ’strong’ (line 424).
Line 425: Does the increased survival rate of nematode necessarily indicate on antioxidant capacity in vivo?
References: To my opinion, major revision should result in a lower number or references for this type of study.
Author Response
Reviewer 2:
This study presents the results on qualitative and quantitative composition of some phytochemicals (mainly flavonoids) in ethanol extracts of fresh marigold flowers (2 cultivars) and the activities of these extracts using several in vitro assays and Caenorhabditis elegans nematode. To my opinion, the value some of these results (phytochemical composition, antioxidant activity) is rather low due to the negligible scientific novelty; while the results with the tested enzymes and worms provide new information on the bioactivities of marigold extract. Therefore, the manuscript, including introduction and discussion, should be better focused by emphasizing new and most important findings. The Authors should consider that numerous articles on T. erecta have been published until now; therefore proving the novelty of the reported findings and their importance should be essential and convincingly shown. The manuscript also contains quite many inaccuracies, some speculative and unclear statements. They are provided below.
Point 1: Line 13 (and elsewhere): Tagetes erecta is also known as Mexican marigold or Aztec marigold (not only an African marigold); therefore, all common names should be given or, preferably, only botanical name should be left in the abstract.
Answer 1: We have removed the common name from the abstract. We have also included both names in the Introduction section and removed from section 3.1.
Pont 2: Line 16: Folin = Folin-Ciocalteu
Answer 2: Corrected
Point 3: Line 23-24: To my opinion, the results of this study do not support the claim ‘T. erecta flowers might be used as functional foods”. More correctly, these results may add some information supporting the possibilities of using this plant for functional food and/or nutraceutical ingredients.
Answer 3: We have substituted the original sentence by the one suggested by the reviewer. (lines 24-25 in the new document).
Point 4: Line 25: To my opinion, ‘functional foods’ is too general and not sufficiently relevant keyword for this study.
Answer 4: We have selected ´polyphenols´ instead of ´functional food´ (Line 26 in the new document)
Point 5: Line 34: I would not agree with the statement: ‘there are few available studies’. For instance, search word ‘Tagetes erecta’ provides a large number of publications in Clavirate Analytics WoS database: Tagetes erecta+antioxidant =80; Tagetes erecta+phenolic composition = 17; Tagetes erecta+protective = 17. In total, thousands of records may be found for different flowers in terms of their composition and bioactivities.
Answer 5: We agree with this observation regarding the number of works which evaluate the composition and certain bioactivities of Tagetes sp. However, the number of works that specifically assess the polyphenolic composition and the in vitro and in vivo bioactivities of fresh flowers are not so numerous. For that reason, in this manuscript we made reference to this concrete type of works. In order to improve the document that sentence has been modified according to reviewer suggestion (Lines 33-34 in the new document).
Point 6: Line 64: ‘Formic acid DPPH”? Most likely, formic acid and DPPH. In addition, DPPH is a stable radical and it should be indicated by the word or, preferably, by the superscript dot.
Answer 6.1: We have written the coma between formic acid and DPPH·.
Answer 6.2: We have substituted “DPPH radical” by DPPH·. in the manuscript; additionally, if DPPH was not preceded by the word “radical” the dot has been inserted as superscript (DPPH·).
Point 7: Line 74: Plant variety is a legal term, following the International Union for the Protection of New Varieties of Plants (UPOV) Convention; in such case more detailed information should be provided for the varieties, e.g. Tagetes erecta L., var? Or these were just different cultivars (accessions)?
Answer 7: The appropriate term is cultivar; the two types of T. erecta included in this study were “orange” and “yellow” cultivar. This information has been modified in the manuscript. The authors used the word variety wrongly due to misunderstanding between English and Spanish.
Point 8: Line 75: How did the Authors ensure that extraction was sufficient for the recovery of bioactive compounds?
Answer 8: The objective of this study was to test the potential of T. erecta in terms of bioactivities and phenolic composition. In order to stablish a relation between the biological activities and the composition of the plant, LC-DAD-ESI/MSn analyses were performed. For the extraction, a protocol using Soxhlet apparatus with ethanol as solvent for 4 h was considered appropriate. Although, cold maceration is usually performed for 24 hours, the Soxhlet extraction allows shortening times due to the heat and the reflux effect.
Point 9: Line 76: How the temperature of 110 C was reached in a Soxhlet using ethanol (boiling point = 78 C)?
Answer 9: We appreciate this comment as we should have been more accurate. The temperature for ethanol during the Soxhlet extraction was not 110 0C. The thermostatic bath was set at 110 0C to provide an efficient heat flow to the flask and ensure reaching the boiling temperature. To be in agreement with that fact we have changed the explanation on the manuscript (Line 73 in the new document).
Point 10: Line 76: Fresh flowers contain large portion of water, which is miscible with ethanol. What was the content of remaining water after solvent removal or water was also evaporated? How it was assessed?
Answer 10: We removed the solvent from the extract mixture until dryness. In fact, the extract showed a powder consistency. We used a Buchi V-710 pump model, which can reach easily pressures between 2 and 6 mbar and delivers 3.1 m3/h. The working temperature for the rotary evaporator bath was 40 0C and this device was working for 3h. After 2 h no liquid was observed in the collecting bottle but we waited one more hour to be completely sure. Taking into account the previous mentioned facts and that under 6 kPa the azeotrope problem for the ethanol+water mixture disappears, we are sure that there was no water in the final extract. Apart from that, the extract was kept at -20 0C and in N2 atmosphere.
Point 11: Line 89: ‘behaviour’ = data.
Answer 11: We have applied this correction.
Point 12: Line 86: The yield of the extract (e.g. from 1 kg of fresh or dried material) in such study would be very important for characterizing the potential of the plant. For instance, the extract may be strong antioxidant but when its yield is very low, the prospects of commercial plant application may be low due to economic issues.
Answer 12: The yield information has been included in section 3.1. (Lines 196-197 in the new document).
Point 13: Line 100: What was the reason of selecting this article for the citation? ‘Folin-Cicolteau’ = Folin-Ciocalteu.
Answer 13: The reason for choosing this reference is because we performed this assay in 96 well microplates with slight modifications.
Point 14: Line 101-102: The same as in line 81?
Answer 14: No, it is not exactly the same concentration. We have changed the explanation to make it clearer.
Point 15: Line 102: ‘Na2CO3’ = Na2CO3.
Answer 15: Corrected
Point 16: Line 105: What was the reason of using pyrogallol, when gallic acid was available?
Answer 16: Results for the Folin-Ciocalteu can be expressed using different phenolic compounds (gallic acid, caffeic acid, etc.); in our case, pyrogallol was chosen because we have this compound in our laboratory to perform certain experiments included in the Spanish Pharmacopoeia in relation with phenolic compounds. Additionally, other manuscripts have expressed total phenolic content as pyrogallol equivalents (Mohajer et al., Phytochemical constituents and radical scavenging properties of Borago officinalis and Malva sylvestris. Industrial Crops and Products 94(2016):673–681
Point 17: Line 109: ‘The radical scavenging activity (RSA) or’ may be deleted.
Answer 17: Corrected.
Point 18: Line 110: This is confusing: DPPH radicals are not reduced ‘through spectrophotometric techniques’ (should be rephrased).
Answer 18: The sentence has been modified to make it more accurate
Point 19: Line 111 (and elsewhere): ‘0,04’ = 0.04
Answer 19: Corrected.
Point 20: Lines 114-117: This is rather trivial and may be replaced by the simple explanation in the text.
Answer 20: Corrected.
Point 21: Line 125: ‘FeSO4·7H20’ = FeSO4·7H20
Answer 21: Corrected.
Point 22: Lines 159, 178 (and elsewhere): Citation style: Is it correct?
Answer 22: Corrected.
Point 23: Line 180: E. Coli = E. coli
Answer 23: Corrected
Point 24: Line 190: E. coli should be in italics
Answer 24: Corrected
Point 25: Data in section 3.1. From the scientific novelty point of view the value of this data is rather low; it just repeat previously reported information on the composition of phenolics in T. erecta (reporting the composition of other samples is just a technical work and provides very little novelty). There is also a question regarding the accuracy of MS (also in Table 1): the Authors provide m/z of pseudomolecular ions [M-H] and fragments as whole numbers (without any decimals), which indicates that the accuracy was 1 amu. Is that correct? Also standard deviations in Table 1 are unbelievably low for biological material. Therefore, it may be supposed that these values are obtained from 3 replicate chromatographic runs of the same extract, which is not sufficient for the reliable statistical evaluation. For this purpose, 3 replicate extractions from the same plant accession had to be performed. In addition, the content of these compounds in the raw plant material was not determined.
Answer 25: In fact, some of the tentatively identified compounds have already been described by other authors and it may seem repetitive. Thus, different compositions were obtained by our identification and Navarro-González et al. (2015). These last authors identified 12 different phenolic compounds, including anthocianin and non-anthocyanin compound, nevertheless, we did not identify anthocyanin compounds. Moreover, in our study we identified 2 laricitrin-di-hexoside and 3 laricitrin-hexoside, where the previous authors only found one laricitrin-di-hexoside and 2 laricitrin-hexoside. We also identified a digallic acid and laricitrin-galloyl-hexoside, and both these compounds were not found in the previous study. As other compounds were found by these authors and not reported in our samples. These differences could be due to a diverse of factors, that are known to affect the chemical composition of plants, such as the timing of harvesting, geographic location/climate and the plant adaptation to the soil conditions. Additionally, differences in the plants genotype may also be associated with chemical variability, therefore affecting its chemical composition. Moreover, these differences could be related to the different extraction procedures applied in both studies, but also to different applied solvents. The extraction methodology highly affects the extraction of compounds, thus it will affect the overall composition in bioactive compounds, and in this case phenolic compounds. Therefore, it is essential to present the phenolic composition of the study extract and the same assumption of the phenolic composition cannot be taken into consideration.
As for the accuracy of the mass spectrum, the equipment has a greater sensitivity, thus in our opinion it is much simpler to understand the fragmentation pattern by rounding the decimals of the m/z values. Thus, the values described in the table result from a rounding of the m/z values, this methodology is used by several authors specialists in the phenolic compounds, with the aim of reducing the difficulty in reading and analyzing the results obtained.
The standard deviations are not, in our opinion, affected by the type of material studied, but dependent on the mean values of 9 replicates, resulting from the analysis, in which the CV % between the mean value and standard deviation should not be greater than 5%, for the result to be reliable. The %CV of Table 1 revealed values ranging from 0.03% to 3.66%, which we consider to be reasonable values.
Regarding the expression of results in mg per g of extract, in this case we do not think it is very important to express the results in raw plant material, because this study is evaluating the bioactive potential of T. erecta extract, therefore a correlation of the compounds found in the extract should be present. Moreover, this expression also depends on the applicability of the study and in this case T. erecta would have the main objective to be used for food or pharmaceutical purposes, not in a direct way, but by using its extract. Therefore, the phenolic compounds and bioactivities studied were all expressed with respect to the extract and not to the dry plant, since the term of comparison would be the extract. Nevertheless, the yield of extraction was added, being easy its conversation in a dry weight bases by the readers.
Point 26: Line 263: ‘antioxidant models’ = assays.
Answer 26 Corrected
Point 27: Lines 262-264: The sentence should be rephrased.
Answer 27: The previous sentence has been rephrased: “In order to confirm antioxidant properties of the extracts, three different methods have been used, namely Folin-Ciocalteu, DPPH· radical scavenging and FRAP assays”
Point 28: Line 264: What means ‘appropriate evaluation’? All these assays are based on a single electron transfer (hydrogen atom transfer may also proceed in DPPH assay) and are not relevant to biological systems. The experts strongly recommend using with FC/DPPH/FRAP assays an ORAC assay, which is better linked to the oxidation processes in biological systems.
Answer 28: We agree with the reviewer suggestion. However, although the reaction is only about transferring an electron from one molecule to another, the exchanged energy depends on the molecules from which the electron migrates; taking into account this consideration, there is a difference among the electron transfer reactions. We will consider other methods including ORAC for future works.
Point 29: Lines 268-269 (the title of Table 2): the units are given in the first row of the Table for every assay; it is not necessary to repeat it in the title.
Answer 29: Corrected.
Point 30: Table 1 and 2: What mean ‘p values’ in these tables? Usually, significant differences between values are indicated by different superscript letters.
Answer 30: p values in table 1 and 2 were calculated in order to detect significant differences between the two cultivars (orange vs yellow).
Point 31: Lines 273-304: To my opinion. this part should be substantially shortened; in general antioxidant capacity measured in this study also is of a little novelty for marigold flowers.
Answer 31: We have taken this suggestion into account, deleting certain sentences.
Point 32: Line 308: The values in Table 2 do not provide information on ‘dose-dependent’ character of their inhibitory activity.
Answer 32: Corrected.
Point 33: Lines 316, 326: Why systematic names in the titles are given using a normal script (not italics)?
Answer 33: Corrected.
Section 3.4: Can these results be compared with: Pina-Vazquez, DM et al.? Anthelmintic effect of Psidium guajava and Tagetes erecta on wild-type and Levamisole-resistant Caenorhabditis elegans strains J. Ethnopharmacol., 202, 92-96 (2017).
Answer 34: No, they cannot be compared because the research of Pina et al. (2017) is focused on locomotion and egg-laying behaviors, while the goal of our experiment is the measurement of the impact of the extracts on the viability on C. elegans.
Point 35: Lines 322-324: To my opinion this comparison is not adequate: essential oils are completely different substances; also the cited study was performed with mice.
Answer 35: We had deleted that comparison.
Point 36: Line 326: The title should be rephrased.
Answer 36: Previous tittle “Positive effect of T. erecta flower in survival rate of C. elegans under lethal oxidative stress.” New suggested tittle “T. erecta exerts protective effects in C. elegans under lethal oxidative stress.”
Point 37: Lines 332-333: Not clear, when compared with the data in Fig.2 showing that survival rate at the highest concentration was several times higher compared with control. Should be rephrased (not ‘by’ but ‘from….to’).
Answer 37: We had modified the paragraph according to reviewer’s suggestion.
Point 38: Line 335: Not clear: ‘The enhanced response to oxidative stress of extracts’.
Answer 38: We had modified the paragraph
Point 39: Line 338: This is a too preliminary and speculative assumption; not supported by this study.
Answer 39: This sentence was eliminated
Point 40: Lines 339-340: This is in disagreement with the statement in lines 298-299.
Answer 40: According to the authors this is not in disagreement due to the fact that lutein may have ferric reducing activity in vitro but may not have in vivo effects against juglone in the C. elegans model.
Point 41: Lines 343-344: Should be rephrased.
Answer 41: Corrected.
Point 42: Line 351: ‘of the flowers’ = of extracts.
Answer 42: Corrected.
Point 43: Line 355: ‘in both extracts’ = for (or ‘in case of’) both extracts.
Answer 43: Corrected.
Point 44: Lines 358-360: This sentence should be rewritten.
Answer 44: We have changed the original sentence: “Among the treated groups, a dose of 250 µg/mL of extract let reach the best lifespan, prolonging the activity for both extracts by a mean lifespan extension of around 18% and 17%, respectively.”
New sentence included: “The best dose prolonging the lifespan, for both extracts, was 250 µg/mL by a mean lifespan extension from 18 days (control group) to 22 days for orange flowers, and from 15 days (control group) to 18 days for yellow one”.
Point 45: Line 371: To my opinion, this figure may be simplified by leaving only the highest doses; the effects of other doses are shortly explained in the text.
Answer 45: Lifespan assay is, in most cases, represented showing all the treatments survival curves against control groups. Therefore, in the original manuscript has been represented in this manner. Nevertheless, we have changed, as suggested, for a better understanding.
Point 46: Line 414: ‘therapeutic strategies’? More correctly, ‘disease preventing or reducing their risks’.
Answer 46: We have changed the sentences to: “The use of functional foods has a great potential to reducing the risk in age-related diseases”
Point 47: Line 415: ‘foods needs’ = foods need.
Answer 47: Corrected
Point 48: Line 419: What mean ‘non-nutrient bioactives’? A nutrient is a general term, encompassing both macro and micronutrients.
Answer 48: Non-nutrient bioactives can be considered as a synonym for “phytochemicals” (= secondary plant metabolites without nutritive value). These compounds are polyphenols, terpenoids or alkaloids, among others. The term nutrient refers to macronutrients (carbohydrates, lipids, proteins) or micronutrients (vitamins, minerals, oligoelements); however, plants contain a wide range of bioactive substances that cannot be considered as nutrients from a nutritional point of view.
Point 49: Line 422: ‘reveals’ or confirms previously reported data?
Answer 49: Corrected.
Point 50: Line 422: When the concentration is considered as ‘high’ (‘medium’, ‘low’)? Also regarding ’strong’ (line 424).
Answer 50: We have deleted the term high and strong to avoid misunderstanding.
Point 51: Line 425: Does the increased survival rate of nematode necessarily indicate on antioxidant capacity in vivo?
Answer 51: An increase in the survival rate of the nematode doesn´t necessarily means an antioxidant in vivo activity. However, we can confirm is that our extract protects against oxidative stress caused by juglone.
Point 52: References: To my opinion, major revision should result in a lower number or references for this type of study.
Answer 52: The authors have followed this recommendation, reducing the total number of references from 55 to 43; the more relevant references were kept.
Round 2
Reviewer 2 Report
The suggested improvements have been made, when it was possible to do without additional experimental work. Although some answers could be further discussed, it may be recognized that the results posses scientific value.